# Machine Learning Applications to Maintain the NuMI Neutrino Beam Quality at Fermilab †

Don Athula Wickremasinghe [1,*] , Yiding Yu [2] , Eduardo A. Ossorio Alfaro [2], Sudeshna Ganguly [1], Katsuya Yonehara [1] and Pavel Snopok [2]

[1] Fermi National Accelerator Laboratory, Batavia, IL 60510, USA; sganguly@fnal.gov (S.G.); yonehara@fnal.gov (K.Y.)

[2] Department of Physics, Illinois Institute of Technology, Chicago, IL 60616, USA; yyu79@hawk.iit.edu (Y.Y.); eossorio@hawk.iit.edu (E.A.O.A.); snopok@gmail.com (P.S.)

\* Correspondence: athula@fnal.gov

† Presented at the 23rd International Workshop on Neutrinos from Accelerators, Salt Lake City, UT, USA, 30 July–31 July 2022.

**Abstract:** The NuMI target facility at Fermilab produces an intense muon neutrino beam for the NOvA (NuMI Off-axis $\nu_e$ Appearance) long baseline neutrino experiment. Three arrays of muon monitors located downstream of the hadron absorber in the NuMI beamline provide the measurements of the primary beam and horn current quality. We have studied the response of muon monitors with the proton beam profile changes and focusing horn current variations. The responses of muon monitors are used to develop machine learning (ML) algorithms to monitor the beam quality. We present the development of the machine learning applications and future plans. This effort is important for future applications such as beam quality assurance, anomaly detection, and neutrino beam systematics studies. Our results demonstrate the advantages of developing useful ML applications that can be leveraged for future beamlines such as LBNF.

**Keywords:** NuMI beamline; machine learning; NuMI neutrino beam

## 1. Introduction

The Neutrinos at the Main Injector (NuMI) beamline [1–3] at the Fermi National Accelerator Laboratory in Illinois has been designed to deliver an intense muon neutrino beam to NuMI neutrino experiments. Protons of 120 GeV from the main injector collide with a fixed graphite target to produce the neutrino beam for experiments. The charged particles produced from the proton interactions with the target nuclei are focused on a 675 m long, 2 m diameter cylindrical decay pipe by using two focusing horn systems which operate with 200 kA horn current. The mesons may decay into neutrinos and muons before they are absorbed through the hadron absorber, located after the decay volume. Some of the high-energy muons produced from the meson decay may pass through muon monitors that are located after the hadron absorber.

The muon monitors are an array of helium gas ionization chambers [4]. Each muon monitor has been built with 81 parallel plate ionization chambers with an electrode spacing of 3 mm. Every charged particle ionizes the helium gas to produce ions and electrons when they pass through the ionizing chambers. Muon monitors are sensitive to the primary beam changes and horn current variations. Unique responses of muon monitors are useful to build machine learning applications to monitor the quality of the NuMI neutrino beam.

After the MINOS detector shut down in February 2019, the muon monitors provide useful information to monitor the beam quality, target health, and horn performance and identify issues with the beamline alignment. Our goal is to improve and monitor the performance of neutrino beam delivery for neutrino experiments by applying modern artificial intelligence and machine learning techniques.

## 2. Muon Monitor Responses

The three muon monitors have a unique response to the proton beam variables. In this paper, we report the muon monitor responses to the proton beam position changes at the target as an example. The data have been recorded by moving the proton beam horizontally and vertically for selected horn current settings. The beam positions at the target have been extrapolated by using two sets of horizontal and vertical beam position monitors (BPMs) upstream of the target.

We report the correlation of the muon monitor centroid measurement as a function of the beam position at the target. The responses of the muon flux centroid to the horizontal and vertical beam variations for 200 kA horn current setting are shown in Figure 1. The correlation of the muon monitor centroid to the proton beam position has been fitted using linear function. The study shows that the three muon monitors are responding differently to the proton beam variations. According to the vertical scan, the non-linear response of the muon flux centroid is visible at the upper and lower limits of the proton beam position.

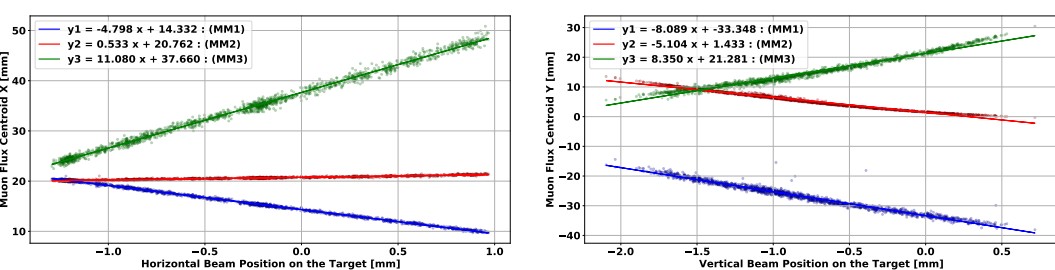

**Figure 1.** Muon flux centroid measurements as a function of proton beam position at the target. Blue, red and green distributions are the measured centroids on muon monitor 1, 2 and 3 respectively. The horizontal and the vertical scans are shown in the **left** and **right** plots, respectively.

## 3. Predicting Beam Parameters with ML

In this study, we present a machine learning approach to predict the proton beam position at the target, beam intensity, and the focusing horn current based on the downstream muon monitor signals. The responses of the individual pixels in the muon monitoring system to the beam and horn current variations have been taken into account to train an ML model.

### 3.1. Data

The data samples have been collected from the spill-by-spill time series measurements of devices in the NuMI beamline for beam settings and horn current settings. The pedestal-subtracted signal measurements of 241 pixels (two dead pixels from muon monitor 1 have not been taken into account for ML applications) of three muon monitors were taken as input variables for the ML model. The randomly sampled training (70%) and validation (30%) data samples were selected from the target scan data collected on 12 December 2019 and a few hours of selected normal operation data.

### 3.2. Machine Learning Method

The ML model architecture is defined as a fully connected multilayered artificial neural network (ANN) with multiple hidden layers. The output of each node in the layers is calculated by an appropriate "activation function". The ANN is designed by using 241 pixel measurements of three muon monitors as input nodes and multiple hidden layers and four output nodes to predict the proton beam position (horizontal and vertical), beam intensity, and horn current. ANN has been optimized by using a hyperparameter tuning process to obtain the optimal architecture. The hyperparameter tuning is carried out by searching the best combination of the number of hidden layers, number of nodes, batch size, learning rate and the activation functions. Figure 2 shows an

example of the number of nodes searched for the first and fourth hidden layers. The plot shows the correlation between the number of nodes in the first and the fourth hidden layers.

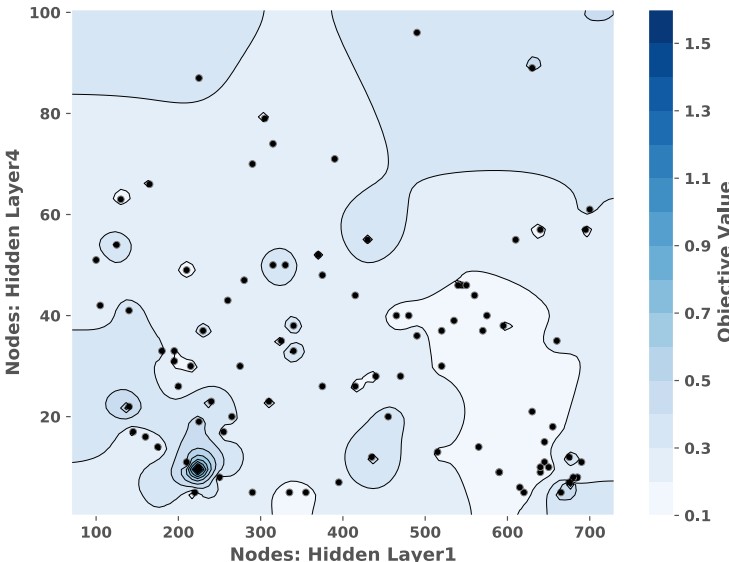

**Figure 2.** The correlation between the number of nodes in the 1st and the 4th hidden layers. Example of the hyperparameter optimization to obtain the number of nodes in the hidden layers. The hyperparameters are tuned based on obtaining the lowest objective values as shown in the contour plot.

The network is trained to obtain the best weights to predict the network output by minimizing an appropriate loss function. The best-optimized ANN structure that we have achieved from our network optimization is described in Table 1 with four hidden layers, a learning rate of $\eta = 10^{-5}$, and a batch size of 32.

**Table 1.** The optimized ANN model with the number of hidden layers, parameters, and the associated activation functions. Each node in the first layer has been connected to 243 inputs from the muon monitor signals and the output layer is predicting the beam position, beam intensity, and the horn current.

| Layer | Shape | Parameters | Activation |
|-------|-------|------------|------------|
| 1 | 480 | 116,160 | tanh |
| 2 | 130 | 62,530 | sigmoid |
| 3 | 135 | 17,685 | sigmoid |
| 4 | 11 | 1496 | sigmoid |
| 5 | 4 | 48 | linear |

A comparison of the model predictions of four output variables on the validation data is shown in Figure 3. The top left and top right plots in Figure 3 show the predictions of the proton beam in horizontal and vertical positions. The beam intensity and the horn current predictions are shown in the bottom left and right plots in Figure 3.

The model has been tested with randomly selected datasets for normal beam operations.

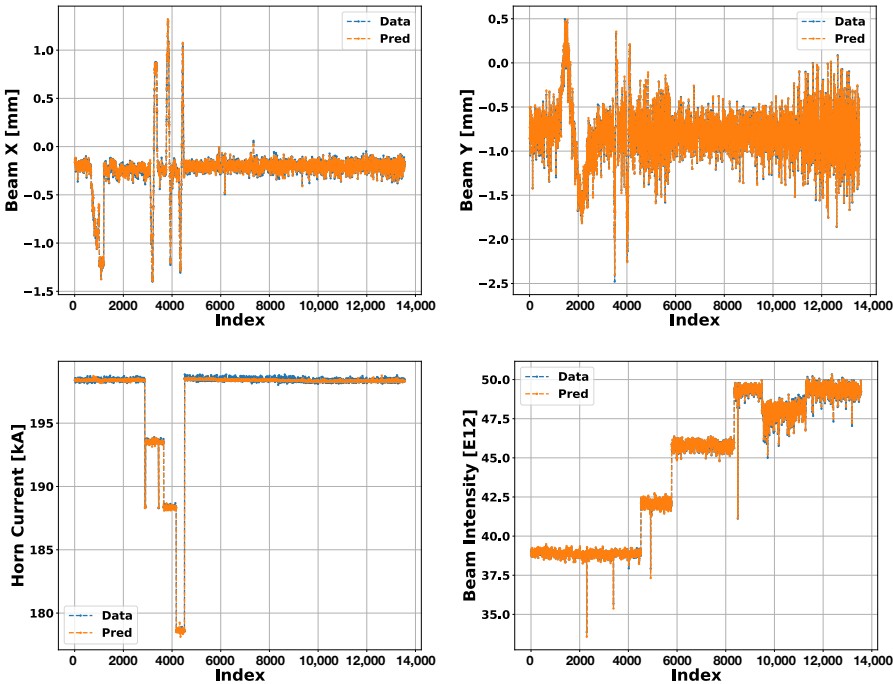

**Figure 3.** A test of the model predictions (orange) of all four output variables for 12 December 2019 target scan data (blue). The beam position is horizontal and vertical; the beam intensity and the horn current are shown in the **top left**, **top right**, **bottom left**, and **bottom right**, respectively.

## 4. Simulation Effort

In this section, we describe the efforts to build ML models by using simulation data. This effort may help to understand some of the rare "anomaly" scenarios such as horn tilt or slip, target tilt, target deterioration, and density effects. The simulation data for ML applications are generated using NuMI beamline simulation (G4NuMI), which is based on geant4 simulation tools [5].

A uniform beam simulation technique is used to generate a significant amount of Monte Carlo data samples for machine learning applications by varying the incident beam parameters and horn current settings. In this technique, we generate a uniformly distributed single simulation data sample for the selected beam variable range. The uniformly distributed sample is then used to generate Gaussian beam profiles for the selected beam positions and widths.

A linear regression model has been tested to predict the horizontal proton beam position and the horn current from simulation data. In this study, we take 243 muon monitor pixels as inputs for the linear model training for each targeted prediction variable. After the training, the model shows a high accuracy of predicting the beam position (left) and the horn current (right), as shown in Figure 4.

Another supervised ML model has been studied by applying convolutional neural network (CNN) techniques. In this study, the model reads each muon monitor pixel information ($9 \times 9$) as an image for the model training. The model reads the image features to make predictions on the targeted output variables. Image information is processed through $3 \times 3$ convolutional filters, a $2 \times 2$ max pooling layer, and a fully connected multilayered perceptron (MLP) during the training process. The results from the model predictions are shown in Figure 5. The top two distributions show the comparison of the beam position predictions along the horizontal and the vertical directions. The bottom plot shows the horn current prediction.

Both models provide a good fit with high prediction accuracy because our simulation data samples describe more ideal conditions than real beam data. Unlike simulation data,

real beam data is noisy and has background effects on measurements from known and unknown sources.

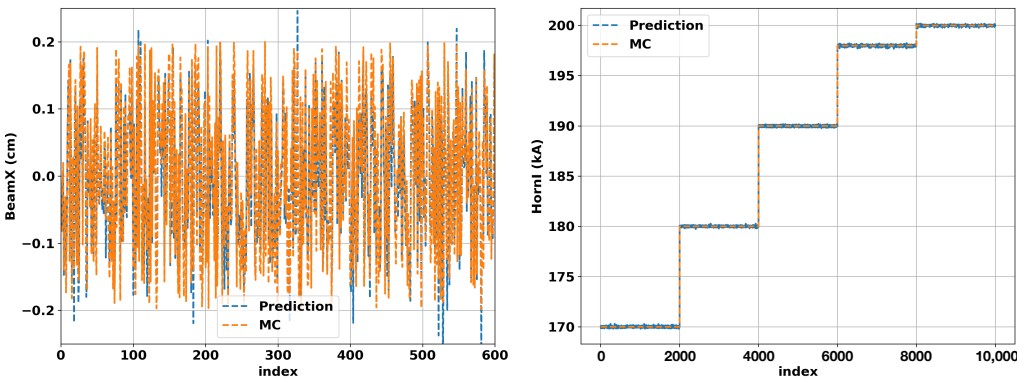

**Figure 4.** Horizontal beam position (**left**) and horn current predictions (**right**) using a linear regression model on simulation data.

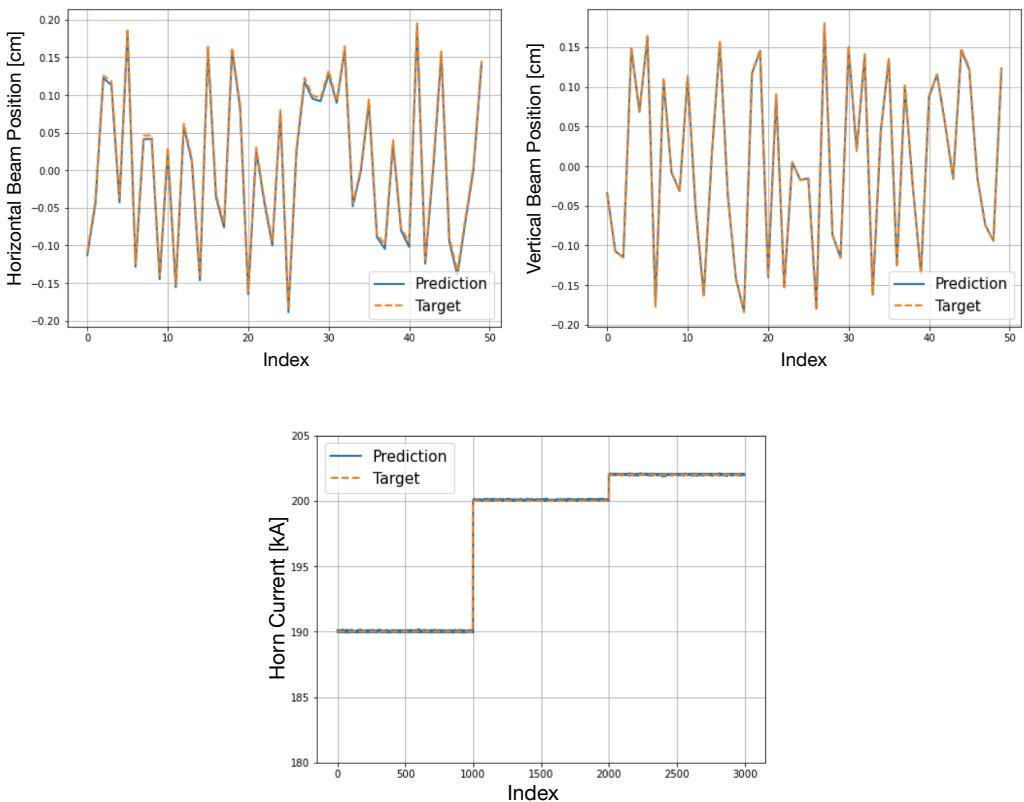

**Figure 5.** Beam position and horn current predictions using CNN model on simulation data. **Top left** and **right** show the horizontal and vertical beam positions. The **bottom** plot compares the horn current predictions.

## 5. Summary and Outlook

In this paper, we summarize the progress of machine learning applications by taking into account the downstream muon monitor signals. These ML predictions give an extra monitoring of the beam and horn current behaviors. This will be helpful for monitoring the beam performance and developing trends or issues during regular beam operations. ML model building with simulation data is useful for predicting rare incidents and anomalies.

**Author Contributions:** Conceptualization, D.A.W., K.Y. and Y.Y.; methodology, D.A.W.; software, D.A.W., Y.Y. and E.A.O.A.; formal analysis, D.A.W., Y.Y. and E.A.O.A.; investigation, D.A.W., Y.Y. and E.A.O.A.; data curation, D.A.W. and Y.Y.; writing—original draft preparation, D.A.W.; writing—review and editing, D.A.W., P.S. and S.G.; visualization, D.A.W. and S.G.; supervision, D.A.W., K.Y. and P.S. All authors have read and agreed to the published version of the manuscript.

**Funding:** This work is supported by the Fermi Research Alliance, LLC manages and operates the Fermi National Accelerator Laboratory pursuant to Contract number DE-AC02-07CH11359 with the United States Department of Energy. This work is partially supported by the U.S. Department of Energy grant DE-SC0019264.

**Institutional Review Board Statement:** FERMILAB-CONF-22-878-AD.

**Informed Consent Statement:** Not applicable.

**Data Availability Statement:** Data is unavailable due to privacy.

**Acknowledgments:** This manuscript has been authored by Fermi Research Alliance, LLC under Contract No. DE-AC02-07CH11359 with the U.S. Department of Energy, Office of Science, Office of High Energy Physics.

**Conflicts of Interest:** The authors declare no conflict of interest.

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
