# Peer review of "Machine Learning Applications to Maintain the NuMI Neutrino Beam Quality at Fermilab†"

_psf, doi:10.3390/psf8010040_

Round 1

Reviewer 1 Report

There are some mistakes in figures. Check Fig1's axis title and description. Check Fig2's axis title.

Why use ANN instead of CNN in real data? How is the ML method better than traditional physics method? How to avoid overfitting in your trainning? 

Author Response

Dear Reviewers,

Thank you for reviewing the paper.  Please see my answers below for your comments and questions. I have attached the updated paper as a PDF file.  

There are some mistakes in the figures. Check Fig1's axis title and description. Check Fig2's axis title.

I have fixed all mistakes. I will upload the latest version of the paper for your consideration. 

Why use ANN instead of CNN in real data? 

First, we built ANN model with real data to see the performance on predicting the beam parameters. That ANN model shows a significantly good prediction accuracy on the validation process. We are continuing the study with exploring other model architectures such as CNN to improve the prediction accuracy. 

How is the ML method better than the traditional physics method? 

It depends on the problem that we are going to solve. 

In our problem, we have to develop a complex and nonlinear physics model by taking account of the horn focussing mechanism with several assumptions and particle scattering through the matter. Building such a model with high prediction accuracy is challenging. 

ML method has the capability to model the predictions by taking account all complex correlations and data patterns. We are able to overcome the complexity of the problem by building a ML model with a large set of training data and also by tuning model parameters. 

How to avoid overfitting in your training?

The training algorithm is designed to exit and save the previous model if the training / validation loss starts to diverge based on the defined loss function. 
